# Scaling Parameter-Efficiency with Distribution Shifts for Domain Adaptation

## Abstract

Distribution shifts between source and target domains pose significant challenges to the generalization capabilities of machine learning models. While foundation models are often fine-tuned to adapt to new domains, their increasing size has led to a rise in the computational resources required for domain adaptation. This has driven interest in Parameter-Efficient Fine-Tuning (PEFT) methods, which have shown strong performance on in-domain tasks. In this work, we investigate how PEFT methods scale with varying degrees of distribution shifts and propose a novel PEFT method designed for domain adaptation. We select an English pre-trained Large Language Model (LLM) as the foundation model and apply PEFT techniques across tasks that progressively introduce larger distribution shifts. Specifically, we begin with SuperGLUE English benchmark, followed by a multilingual inference task for high-resource and low-resource languages, then a multimodal image captioning task. Finally, We introduce a novel multimodal and multitemporal radar interferometry task for detecting charcoal production sites in remote areas. Separately, we propose a PEFT method that augments matrix vector products with learnable parameters, inducing a learning paradigm that conditions on both training data and encoded information. Our method is competitive against SOTA PEFT methods for English tasks and out-performs SOTA methods for larger distribution shifts i.e. low-resource multilingual, image captioning, and radar interferometry tasks.

## 1 Introduction

Foundation Models are increasingly becoming prevalent in research and industry applications. As performance demands increase, training foundation models increasingly require vast amounts of high-quality data and significant computational resources Kaplan et al. (2020); Xu et al. (2025). Transfer learningZhuang et al. (2020) enables the adaptation of pre-trained foundation models to new tasks using fewer examples compared to training from scratch. This accelerates the application of AI in low-resource environments, reducing the need for extensive data and computation.

As Foundation Models grow in size, Transfer Learning becomes more resource intensive. This has motivated research on efficient fine-tuning approaches. We can classify these efforts as either (1) Numerical Precision methods or (2) Parameter Efficient Fine-Tuning (PEFT) methods. Numerical Precision methods focus on reducing the precision of the parameters to save on compute. By preferring low numerical precision, such as FP16 over FP32, training can be $4\times$ faster on GPUs Micikevicius et al. (2017); Xu et al. (2025). A widely adopted method for this approach is DeepspeedRasley et al. (2020). In addition to using low numerical precision, Deepspeed also employs parameter offloading from GPUs to CPU during training. By combining these 2 methods, researchers have been able to train/fine-tune large models on small GPUs.

PEFT methods save on compute by only fine-tuning a few parameters, leaving the rest of the model frozen. This reduces the gradient computation and storage operations, saving both compute and GPU memory. A widely adopted PEFT method is Low Rank Adapter (LORA)Hu et al. (2021) which focuses on only updating the low ranks of the parameter matrices. This has resulted in the optimization of less than 1% of large models, performing similar to the fine-tuning of full parameters. In practice, both Deepspeed and PEFT are used together.

Distribution shift between training and target task data is at the root of the out-of-distribution problems Yang et al. (2024). Currently deployed foundation models Brown et al. (2020); Achiam et al. (2023); Touvron et al. (2023) are trained on all available online data, resulting in zero-shot capabilities. However, transfer learning is still required for never seen data and custom use cases. This reality, combined with the size and complexity of modern foundation models, motivates the need to investigate how PEFT methods scale with distribution shifts.

We hypothesize that significant distribution shifts hinder transfer learning and pose challenges for PEFT methods. In this study, we investigate how PEFT techniques perform under varying degrees of distribution shift. Using a Large Language Model (LLM) pretrained on English as our foundation model, we apply PEFT across a range of tasks with increasing distribution divergence. We begin with SuperGLUE Wang et al. (2019), a benchmark for English-language tasks, followed by a multilingual natural language inference task Conneau et al. (2018a). To explore more extreme distribution shifts, we then evaluate on a multimodal image captioning task Lin et al. (2014) and conclude with a remote sensing timeseries classification task, which we frame as both multimodal and multitemporal.

In this paper, we show that, indeed, current PEFT methods do not scale with distribution shifts. We propose a novel PEFT approach that conditions on pre-trained weights, augmenting knowledge by matrix vector product. Finally, we introduce a novel remote sensing application for Synthetic Aperture Radar (SAR) imagery, leveraging on interferometry to create a timeseries of earth surface vertical displacements which we use to predict the location of charcoal production kilns. Our proposed PEFT method is competitive on the English and high-resource language benchmarks, and outperforms other PEFT methods on the multimodal, multitemporal and low-resource language benchmarks.

## 2 BACKGROUND AND RELATED WORK

### 2.1 DOMAIN ADAPTATION

Classical machine learning is based on the assumption that the dataset in question is independent and identically distributed (i.i.d.) Murphy (2012). Therefore, we can expect relationships learnt by parameters to generalize to unseen data. In real life, the i.i.d. assumption does not always hold, motivating methods for domain adaption Farahani et al. (2021). One way to address this challenge is to reduce the chances of non-conformity to the i.i.d. assumption. Large Language Models (LLMs) implement this by training on the vast data available on the public web, finetuning on a specific task in just a few optimization steps. However, domain adaptation questions still exist for LLMs with out-of-distribution data as it happens in multimodality, multilinguality etc. This can be solved by training from scratch but requires extensive compute resources, thus motivating the study of efficient methods. In this work, we investigate how PEFT methods scale with distribution shifts to determine whether they can be used to perform efficient domain adaptation.

### 2.2 PARAMETER EFFICIENT FINETUNING

The size of foundation models is increasing faster than hardware advancements. This has motivated research into more efficient finetuning methods. The principle is that, foundation models already encode some level of knowledge and optimizing only a few parameters can induce better performance while saving on compute. Adapter Rebuffi et al. (2017); Houlsby et al. (2019); Liu et al. (2022); Zhang et al. (2023) methods introduce extra trainable parameters to the layers of a frozen pretrained model to reduce memory usage and speed up training. Sparse methods Guo et al. (2020); Zaken et al. (2021); Sung et al. (2021); He et al. (2024) select a subset of the existing parameters for finetuning; for example, BitFit Zaken et al. (2021) only updates the bias weight of neural network layers. Low-rank adaptation methods Hu et al. (2021); Liu et al. (2024); Dettmers et al. (2023) update low-dimension matrices of the model parameters. Input methods Li & Liang (2021); Lester et al. (2021) concatenate optimizable vectors to model inputs, adapting neural activations to perform different tasks. Our method introduces a narrow parallel network to the model, interacting with frozen methods via matrix vector multiplication. During optimization, the new parameters are updated as partial derivatives w.r.t. both encoded knowledge and model inputs. Thus, our method

is an adapter method designed to create capacity for encoding new information while depending on previously encoded information.

## 2.3 SAR INTERFEROMETRY

Synthetic Aperture Radar (SAR) uses signal processing to simulate a large aperture for radar imagery Ramakrishnan et al. (2002). The larger the aperture, the higher the resolution. Synthetic apertures capture high-resolution images using small physical antennas onboard moving aircrafts Stimson (1998). Interferometry Gens & Van Genderen (1996); Rocca et al. (2000) uses two SAR images to measure vertical displacement on the earth's surface with millimeter precision. It uses information from phase differences of the SAR images, orbit information, and earth's curvature to calculate vertical displacement. The quality of the displacement data is termed coherence and is dependent on how fast changes on the surface occur. Low coherence implies that the surface changed too fast, e.g. due to rapid vegetation changes, and displacement values cannot be trusted. SAR interferometry has many applications such as forest biomass prediction Flores-Anderson et al. (2019), flood detection Wu et al. (2023), measuring the subsidence of citiesDelgado Blasco et al. (2019) and monitoring public infrastructure Tarighat et al. (2021); Macchiarulo et al. (2022). In this work, we use SAR interferometry to detect charcoal production kilns. We obtain SAR images recorded by the Sentinel 1A satellite and perform interferometry using the SNAP software. We do not discard low coherence values to avoid gaps in the time series. We will explore increasing the confidence of low-coherence data in future work.

## 3 MATRIX VECTOR PRODUCT AUGMENTATION

Parameter Augmentation is an introduction of a narrow parallel neural architecture to an existing architecture. By freezing the previous architecture's parameters and only optimizing the narrow parallel architecture, it is able to represent new information conditioned on both the inputs and the pretrained parameters. The intuition behind it is that by freezing the pretrained parameters, knowledge is preserved and new free parameters encode new information as a function of both inputs and the previous information. Our empirical results show that this can be applied to use cases ranging from mitigating cross-domain, multilingual and multimodal adaptation, etc. By keeping the augmentation small, we can train less than 1% of the pretrained model.

In this work, we explore whether a small enough augmentation $d_{aug}$ will result in performance comparable or superior to full parameter finetuning. Consider an LLM with a hidden dimension size $d_{llm}$ of 3200, augmenting with a width of 2 means the parameter matrices will be extended with 2 columns, resulting in finetuning 0.06% of the LLM.

To augment the LLM, we concatenate the pretrained and augmenting parameters to form one matrix. Specifically, for each pretrained LLM parameter matrix $W^{llm} \in \mathbf{R}^{d_{llm} \times d_{llm}}$, we create augmented parameter $\hat{W}$ as $\hat{W} = (W^{llm}|W^{aug})$, where $W^{aug} \in \mathbf{R}^{d_{llm} \times d_{aug}}$. All subsequent operations (attention, normalization, feed-forward) are therefore between inputs and augmented weights $\hat{W}$. This preserves the number of matrix operations since we only change the dimensionality.

During full parameter finetuning, hidden state $h_i \in \mathbf{R}^{d_{llm}}$ and its derivative is given by equation 1.

$$h^{llm} = \sum_{j=1}^{d_{llm}} x_j W_{ji}^{llm} \implies \frac{\delta h_i^{llm}}{\delta W_{ji}^{llm}} = x_j + \nabla W_{ji}^{llm} \tag{1}$$

Finetuning a separate augmentation hidden state $h_i^{aug}$ and its derivative can be represented by equation 2.

$$h^{aug} = \sum_{j=1}^{d_{aug}} x_j W_{ji}^{aug} \implies \frac{\delta h_i^{aug}}{\delta W_{ji}^{aug}} = x_j + \nabla W_{ji}^{aug} \tag{2}$$

We can then get the augmented hidden state $\hat{h}$ by a simple concatenation i.e. $\hat{h} = (h^{llm}|h^{aug})$. However, keeping them separate in this manner means that we optimize the augmented parameters dependent on the inputs but independent of the pretrained parameters. Considering the goal is optimizing the augmented parameters conditioned on both inputs and pretrained parameters, we want

to introduce the pretrained parameters into the gradients of the augmented parameters by combining the 2 equations above into the same matrix operation by multiplying an augmented input with the augmented matrix $\hat{W} \in \mathbf{R}^{d_{llm} \times (d_{llm} + d_{aug})}$

$$\hat{h} = \sum_{j=1}^{d_{lmm}} x_j W_{ji}^{llm} + \sum_{k=d_{llm}+1}^{d_{llm}+d_{aug}} x_k W_{ki}^{aug} \implies \frac{\delta \hat{h}}{\delta W_i^{aug}} = x + \nabla W_i^{aug} \tag{3}$$

The derivative in equation 3 still does not have a term for pretrained parameters $W^{llm}$. Considering that LLMs have several layers whose inputs depend on the previous layer, the constant $x$ in the derivative is actually input from the previous layer, thus introducing the pre-trained parameters into the gradient. Say an LLM has $T$ layers, at any layer $t$, we will have a term from the previous layer $t-1$

$$\frac{\delta \hat{h}}{W_i^{t_{aug}}} = \underbrace{\sum_{j=1}^{d_{lmm}} x_j^{t-1} W_{ji}^{t_{llm}-1} + \sum_{k=d_{llm}+1}^{d_{llm}+d_{aug}} x_k W_{ki}^{t_{aug}-1}}_{x^t} + \nabla W_i^{t_{aug}} \tag{4}$$

In summary, matrix vector product augmentation:

1. Preserves the number of matrix operations because the changes only affect the dimensionality of the matrices. Therefore, we can minimize the performance overheads using established methods for accelerating linear algebra computations.

2. During matrix operations, the frozen parameters interact with free parameters within the same matrix products, constraining the free parameters to encode additional knowledge as a function of both the frozen parameters and model inputs.

3. By keeping the size of augmentation small, we can optimize $< 1\%$ of the pretrained model, similar to established PEFT methods.

## 4 SUPPORTING MULTIMODALITY

Multimodal data is an extreme case of distribution shifts, yet rich in potential applications. For uni-modal text tasks, it is sufficient to augment the embedding look-up table. We achieve this by introducing a narrow trainable embedding lookup table to provide augmenting embeddings $x_\theta \in \mathbb{R}^{d_{aug}}$. Concatenating frozen pretrained embeddings, $x$, and augmenting embeddings i.e. $\mathbf{x} = (x|x_\theta)$, $\mathbf{x} \in \mathbb{R}^{d_{llm}+d_{aug}}$ makes input embeddings compatible with augmented matrix vector products in subsequent architecture layers.

To support text-image modalities, we vectorize the images using CLIP Radford et al. (2021) then use a trainable linear layer, $h_\phi$, to map from CLIP vector dimensions to LLM embedding dimensions. This however does not guarantee that the image embeddings semantics are transferable to the frozen embedding look-up table $tab_{llm}$. To resolve this, we use an anchor token, <image>, which we bias elementwise as shown below.

$$\hat{x}_{image} = h_\phi(x_{image}) \oplus tab_{llm}(< image >) \quad , \quad \hat{x}_{image} \in \mathbb{R}^{d_{llm}} \tag{5}$$

## 5 CASE STUDY: CHARCOAL KILN DETECTION USING SAR INTERFEROMETRY

In this section, we discuss how we use SAR satellite imagery to predict the presence of charcoal production kilns in Somalia. Compared to optical images, radar is robust to inclement weather and lighting variations, making it possible to collect data during heavy cloud cover and low lighting Lu (2007); Soergel (2010). Furthermore, we can leverage on interferometry Bamler & Hartl (1998); Rosen et al. (2000) to add a third dimension to the data. Interferometry uses phase difference of the radar signal to determine changes in vertical displacements on the earth's surface with millimeter precision. This provides a physical metric to analyze activity that disturbs the earth's surface using the same sensor data.

## 5.1 DATA COLLECTION

Charcoal production involves the burial of wood in kilns on the ground, creating vertical displacements on the earth's surface that can be measured using radar interferometry. We compile 500 charcoal sites detected between May and December 2019 from the FAO SWALIM charcoal dataset (Verhegghen et al., 2023). The dataset is set in southern Somalia. It consists of GPS locations and dates for when charcoal kiln sites were detected by humans using high-resolution aerial images.

We compile SAR images recorded by Sentinel 1 Satellite for the same period as the charcoal dataset. The SAR images have a spatial resolution of $14$ meters (i.e. every pixel translates to $14\text{m}^2$ on the ground), and a temporal cadence of 2 weeks. We perform SAR Interferometry process using the images as discussed by Yagüe-Martínez et al. (2016) using SNAP toolbox, creating a time series of vertical displacements for each image pixel. Each time series has 17 data points.

SAR images have spatial localization attributes, i.e., pixels close to each other will tend to have similar characteristics. This phenomenon is caused by the fact that neighboring points on the earth's surface tend to have the same vegetation and terrain which result in similar radar signal backscatter. Therefore, it is more difficult to differentiate between locations that are close to each other compared to locations that are farther apart. To capture this difficulty in out dataset, we sample negative labels for charcoal kiln sites from regions that are outside the GPS polygons but within a 40 meter distance from the closest point of the polygon. Positive labels, on the other hand, are any pixels that are completely surrounded or fall on the edge of a polygon. The ratio of positive to negative labels is 1:2, we will oversample the positive labels during training and weight the classification metrics when evaluating performance.

## 5.2 SUPPORTING MULTIMODALITY AND MULTITEMPORALITY

SAR interferometry over expansive regions is a computation-intensive process. To reduce computation overheads, we will train a model to approximate the interferometry displacement when provided with a pair of SAR image data. We will use SAR images covering 5k square kilometers of northern Uganda to train our interferometry approximation model and apply it to southern Somalia SAR images.

To extract radar image embeddings, we decompose radar pixels to constituent components i.e. amplitude, phase and intensity which we vectorize by concatenating subsequent radar sensor measurements. This ensures that each vector contains the raw data needed to perform interferometry and compute the vertical displacement. We then train a transformer encoder and a regression layer to map the raw radar vectors to the interferometry displacement by minimizing the MSE loss $\frac{1}{n}\sum(\hat{y}_i - y_i)^2$, where $y_i$ is the true vertical displacement computed using SNAP tool. We discard the regression module to obtain a model that takes raw sensor data as input and computes embeddings which can be used to approximate the vertical displacement, similar to how CLIP Radford et al. (2021) computes image embeddings which are used for further processing. We then use a linear layer to map the radar embeddings to the LLM dimension, similar to the text-image modality.

We will interleave the radar embeddings for each timeseries data point with text embeddings to achieve a multimodal and multi-temporal setup. During training, we will only optimize the linear mapper and augmenting parameters. However, the statistical properties of the radar imagery embeddings $x^r$ do not match the token embeddings $x^t$, which creates a need to normalize the radar embeddings. To this end, we adapt the RMS Norm Zhang & Sennrich (2019) by introducing learnable gates $\lambda_\theta^t$ and $\lambda_\theta^r$ to dynamically induce the statics of the mapped radar image vectors to token vectors as shown below. The symbols $\odot$ and $\oplus$ indicate elementwise multiplication and addition.

$$\hat{x}^r = x^r \odot \mathbf{W}_\theta^{rms} \odot \left[ \lambda_\theta^t \cdot \left( \sqrt{\mu((x^t)^2)} \right)^{-1} \oplus \lambda_\theta^r \cdot \left( \sqrt{\mu((x^r)^2)} \right)^{-1} \right] \quad , \quad x^r = h_\phi(x_{radar}) \quad (6)$$

## 6 EXPERIMENT SETUP

To investigate how PEFT methods scale with data from different distributions, we need to select datasets that progressively differ from pretraining dataset. Modern models Raffel et al. (2020);

Zhang et al. (2023); Touvron et al. (2023); Jiang et al. (2023) are trained on publicly available data. This also includes widely used benchmarks SuperGLUE Wang et al. (2019). To guarantee that we can control for this requirement, we select Open-LLaMA v2 Geng & Liu (2023); Touvron et al. (2023) as our foundation model. Open-LLaMA is an open-source LLM trained on open-source English data Computer (2023); Kocetkov et al. (2022); Penedo et al. (2023). Therefore, selecting English, multilingual, multimodal and multitemporal datasets fulfill the criteria for different data distributions.

We test PEFT methods on SuperGLUE English benchmark, XNLI Conneau et al. (2018b) multilingual inference task, MS-COCO Lin et al. (2014) image captioning task and our SAR interferometry timeseries classification task. For SuperGLUE and XNLI, we obtain GPT-style prompts using PromptSource Bach et al. (2022). For remote sensing task, we use the prompt template:

```
Time series data:  1:a, 2:b, 3:c, ..., 17:q
Does the time series data represent a site or not?  Answer with
yes or no.
```

The characters `a, b, c, ..., q` are used as placeholders for radar time series data. Each character embedding is replaced with radar embeddings before the first transformer layer.

All our experiments use Open-LLaMa 3B. We finetune the respective PEFT methods at half-precision floating point numbers Micikevicius et al. (2017); Narang et al. (2017) on A100 GPUs. For LORA and AdaLORA, we set $r = 8$ and $\alpha = 8$. For each task, we perform an independent hyperparameter search for learning rate (between $1e-5$ & $1e-1$), maximum token length and batch size. We finetune PEFT parameters for 1000 iterations if the batched data is insufficient, otherwise we finetune for 3 epochs. Optimization is performed using AdamW optimizer with 20% warm-up rate and cosine learning rate decay. Our method is abbreviated as AUG_X where X is the number of columns in the augmenting matrix.

# 7 RESULTS AND DISCUSSION

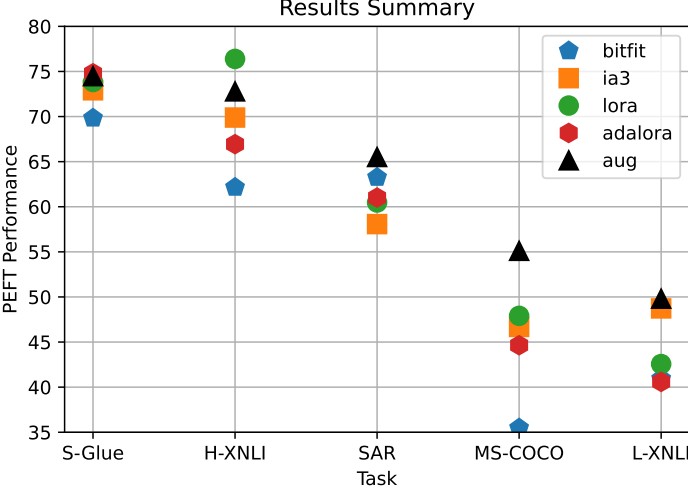

Figure 1: Showing performance summary of PEFT methods across various tasks. Y-axis is normalized to the range (0,100). S-GLUE is super-glue natural language tasks. H-XNLI is high resource langauges XNLI task. SAR is SAR Interferometry timeseries classification task. MS-COCO is multimodal text generation task. L-XNLI is low resource languages XNLI task. Our method (aug) outperforms other PEFT methods on low-resource languages and multimodal tasks while still being competitive for natural language tasks.

## 7.1 ENGLISH BENCHMARK RESULTS

Table 1 shows the performance of PEFT methods on SuperGLUE English benchmark. We add the results of full parameter finetuning for reference on the bottom row. We observe relatively lower performance on Winogrande dataset; an adversarially designed dataset for models. The aggregate

results indicate that our method (AUG_4) and AdaLORA outperformed other PEFT methods and full parameter finetuning, which justifies the case for parameter-efficient learning. The Fourier Transform Gao et al. (2024) method did not perform well in our setup. Furthermore, the method took approximately ×6 the average time it took to train and evaluate the other PEFT methods. For these reasons, we do not include it in the rest of the experiments.

| PEFT Method | BoolQ | Copa | RTE | SST2 | WiC | Wino | Average |
|---|---|---|---|---|---|---|---|
| BitFit | 75.04 | 79.82 | 65.34 | 93.23 | 54.85 | 50.73 | 69.83 |
| IA3 | 76.33 | 78.27 | 74.72 | 92.20 | **63.16** | **52.78** | 72.91 |
| LORA | 79.35 | 78.93 | 77.61 | **94.72** | 60.18 | 52.17 | 73.83 |
| AdaLORA | 77.52 | **83.90** | 79.42 | 93.80 | 61.59 | 52.57 | **74.80** |
| Fourier T | 65.99 | 70.76 | 55.95 | 52.06 | 55.17 | 47.65 | 57.93 |
| AUG_2 | 79.63 | 82.88 | 79.42 | 94.26 | 57.99 | 51.59 | 74.29 |
| AUG_4 | **81.43** | 80.19 | **80.50** | 93.92 | 58.93 | 51.89 | 74.48 |
| Full FT | 83.88 | 81.54 | 77.25 | 94.26 | 57.83 | 48.23 | 73.83 |

Table 1: Showing the performance of PEFT methods on SuperGLUE English Benchmark. Wino column shows results from Winogrande dataset. Except for Copa and Winogrande datasets, performance is computed as accuracy (%) of token generation. For Copa and Winogrande performance is Rouge Score. Our method (AUG_4) outperforms full parameter finetuning.

## 7.2 MULTILINGUAL BENCHMARK

Table 2 shows performance on the XNLI benchmark. Data entries in this benchmark contain English and a second language e.g. French dataset contains data entries where English is mixed with French. Turkish, Swahili and Urdu are low-resource languages. We observe a notable performance drop when comparing high and low-resource languages. This is because of the lexicon overlap between English and the high-resource languages tested. Bitfit consistently struggles with this benchmark. Conceptually, low-rank methods are not designed for encoding new information. Adapter methods, on the other hand, have capacity for knowledge augmentation because they introduce new parameters. This is consistent with the significant drop in LORA performance when crossing from high to low-resource languages. Our method is competitive on high-resource languages and performs better than other PEFT methods on low-resource languages.

| PEFT Method | High Resource Languages | | | Low Resource Languages | | | |
|---|---|---|---|---|---|---|---|
| | French | Deutch | Spanish | Turkish | Swahili | Urdu | Average |
| BitFit | 60.57 | 57.22 | 68.70 | 45.34 | 43.63 | 33.83 | 51.55 |
| IA3 | 69.58 | 66.56 | 73.55 | 55.62 | 47.16 | **43.41** | 59.31 |
| LORA | **76.02** | **75.12** | **78.04** | 44.55 | 45.82 | 37.32 | 59.48 |
| AdaLORA | 67.08 | 64.15 | 69.60 | 43.43 | 42.59 | 35.72 | 53.76 |
| AUG_2 | 72.25 | 69.72 | 61.71 | 49.94 | 47.10 | 37.52 | 56.37 |
| AUG_4 | 71.31 | 70.59 | 76.48 | **57.66** | **50.55** | 41.23 | **61.31** |
| Full FT | 73.89 | 78.68 | 81.71 | 65.08 | 64.33 | 50.47 | 69.09 |

Table 2: Showing the performance of PEFT methods on XNLI high resource and low resource languages. Performance metric is NLI accuracy for all languages. Our method (AUG_4) outperforms other PEFT methods when aggregated across all cross-lingual settings.

## 7.3 MULTIMODAL BENCHMARK (MS-COCO CAPTION GENERATION)

During training, vectorized images are appended to the beginning of the inputs, followed by the prompt text and finally the image captions i.e `<image> image captions are: <image captions>`. During inference, models generate the image caption. Performance is computed by comparing the generated image caption with the list of possible image captions using the Rouge-1 metric. We use the maximum rouge score to aggregate across the dataset.

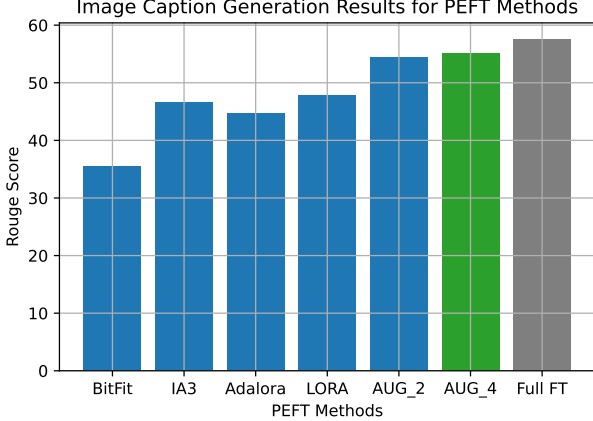

Figure 2: Showing the image caption generation results for the MS-COCO dataset. Performance is measured using Rouge-1 metric. Full parameter finetuning is added for comparison but greyed out. Our method (in green) generated better image captions compared to other PEFT methods.

Figure 2 Shows the results of MS-COCO image caption generation when PEFT methods are used to adapt a language model for image caption text generation task. Our method generates better image captions compared to other PEFT methods and is comparable to full parameter finetuning.

## 7.4 Multimodal Multitemporal (SAR Interferometry Timeries) Classification

We introduce our proposed dynamic normalization between the linear mapper and the LLM to test the effect of dynamic normalization across all PEFT methods. We observe that performance improved for all adapter methods and full finetuning when SAR timeseries data statistics are normalized to text data statistics. However, low-rank methods experience performance decline, see Figure 3. We hypothesize that normalizing other modality data to be similar to text data makes it indistinguishable in the low-rank space, confounding learning for low-rank methods.

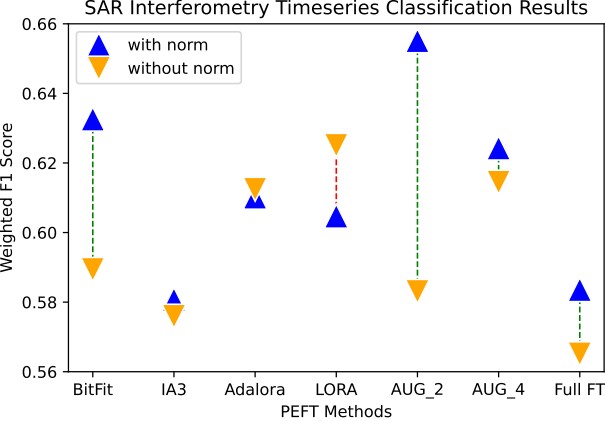

Figure 3: Classification results for SAR interferometry timeseries data. Labels for classification indicate presence or absence of a charcoal production kilns. Green vertical lines indicates that normalization resulted in better classification performance, while red vertical lines indicate classification performance decline after introducing normalization. Our method (AUG_2) achieved the highest weighted F1 Score.

Sentinel 1 SAR images are recorded using a C-band radar sensor. The Frequencies used by C-band radar do not penetrate vegetation, resulting in noise that may inhibit object detection. Separately, high-density regions have similar SAR signals making it difficult to distinguish timeseries data based on physically close locations. To detect charcoal kilns, PEFT methods will have to learn models that generalize across varying vegetation cover, terrain and data density. Figure 4 shows how the classification of timeseries data varies under different conditions. Full parameter finetuning resulted

in significant false positives. Our method (AUG_2), LORA (without normalization) and Bitfit show robustness to the data imbalance and generalized across different radar conditions.

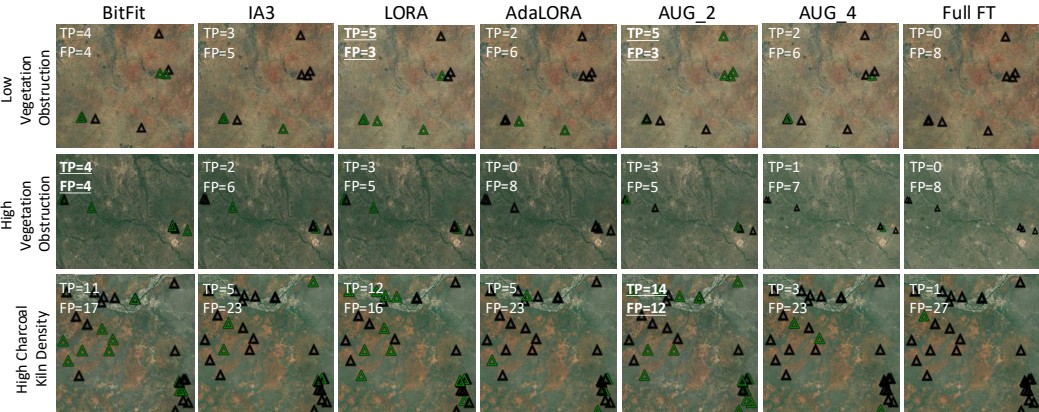

Figure 4: Showing SAR interferometry timeseries classification under different conditions. Green markers indicate true positives while black markers indicate false positives. The first row shows low vegetation-cover scenario, which translates to minimal SAR obstruction. The second row shows high vegetation cover which translates to higher SAR obstruction. The final row shows high charcoal-kiln density implying high charcoal burning activity. The best results for each row are underlined.

## 7.5 PEFT COMPUTATION RESOURCES

| PEFT Method | Tuned Params (%) | Training Memory |
|---|---|---|
| BitFit | 0.0261 | **27.8** |
| IA3 | **0.0163** | 35.7 |
| LORA | 0.1164 | 30.1 |
| ADALORA | 0.1331 | 30.6 |
| AUG_2 | 0.0624 | 39.1 |
| AUG_4 | 0.1248 | 39.1 |

Table 3: Showing the resources expended by PEFT methods. Percentage of training parameters is in reference to number of parameters optimized during full model finetuning. Training memory is GB of GPU memory.

Table 3 shows the compute resources used by PEFT methods. All methods have a number of floating-point operations in the range of $1.68e10$ FLOPs, but we observe differences in GPU memory. All methods result in finetuning less than $1\%$ of the model parameters. IA3 Liu et al. (2022) had the fewest proportion of tuned parameters compared to full model tuning but still required more memory than low-rank methods. Bitfit Zaken et al. (2021) uses the least GPU memory and has the fastest training and inference speeds. Our method used the most memory resources among the PEFT methods. This is because the frozen parameters are still referenced in the gradient computation graph i.e. gradient updates are conditioned on both the inputs and the frozen parameters. IA3, which is also an adapter method, is affected by this phenomenon but limited to only the attention modules. This is a known limitation of adapter methods i.e. the introduced parameters result in small but non-negligible increase in computational costs Liu et al. (2022).

## 8 CONCLUSION

In this work, we examine how the performance of PEFT methods varies across tasks that differ in their distributional shift from English natural language. Our findings reveal that PEFT methods generally do not scale well under increasing distribution shifts, with low-rank approaches being more susceptible than adapter-based methods. We also introduce a novel multimodal and multitemporal task: SAR timeseries classification for detecting charcoal kilns using remote sensing data. Our approach outperforms full-parameter finetuning on both the SuperGLUE benchmark and the SAR classification task, and achieves comparable performance on image captioning. Overall, our method consistently outperforms other PEFT techniques, particularly as task distributions diverge further from English.

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

# A  APPENDIX

