# OpenReview forum: "Scaling Parameter-Efficiency with Distribution Shifts for Domain Adaptation"
_ICLR.cc/2026/Conference — ICLR 2026 Conference Withdrawn Submission_

### Official Review · Reviewer_4P2R · 2025-10-25

**Soundness:** 3
**Presentation:** 3
**Contribution:** 2
**Rating:** 4
**Confidence:** 5

**Summary:**

This paper investigates the performance of Parameter-Efficient Fine-Tuning (PEFT) methods under increasing distribution shifts across tasks. We introduce a novel PEFT technique, AUG, which augments matrix-vector products with learnable parameters conditioned on both the input data and pretrained weights.

The efficacy of AUG is evaluated across a diverse set of tasks: English language understanding (SuperGLUE), multilingual classification (XNLI), multimodal processing (MS-COCO), and multitemporal radar interferometry. AUG is shown to consistently match or outperform existing PEFT methods, demonstrating particular strengths in low-resource and highly distribution-shifted settings.

This paper proposes a new Synthetic Aperture Radar (SAR) imaging task for detecting charcoal kilns, demonstrating that AUG is scalable, efficient, and highly adaptable across different modalities.

**Strengths:**

## Originality
This paper introduces a novel PEFT method—Matrix Vector Product Augmentation (AUG)—that conditions learnable parameters on both input data and pretrained weights.
AUG extends existing PEFT methods such as LoRA, BitFit, and IA3.
It creatively combines adapter-style augmentation with gradient conditioning, and introduces dynamic normalization for multimodal inputs.
Additionally, the application of PEFT to a new domain—SAR interferometry for charcoal kiln detection—is original and expands the scope of PEFT research beyond conventional NLP and vision tasks.

## Quality
The current experimental design is methodical and spans a diverse set of tasks with increasing distribution shifts: English (SuperGLUE), multilingual (XNLI), multimodal (MS-COCO), and multitemporal (SAR).
This comparisons are made against multiple strong PEFT baselines, and the results are presented clearly with appropriate metrics. The mathematical formulation of AUG is well-structured and logically sound.
While the base model used (OpenLLaMA v2) is now not state-of-the-art,
the experiments are reproducible and the methodology is robust.

## Clarity
This paper is clearly written, with a logical flow from motivation to method, experiments, and results. The figures and tables are informative and well-labeled. The mathematical derivations are concise and accessible, and the rationale behind design choices is explained. The multimodal integration and SAR task setup are described in sufficient detail to understand the novelty and implementation.

## Significance
The current work addresses a practical and underexplored challenge: how PEFT methods scale under distribution shifts. This proposed method shows strong performance in low-resource and out-of-distribution settings, which are critical for real-world deployment.
The introduction of a new multimodal and multitemporal task (SAR-based classification) demonstrates the broader applicability of PEFT beyond language and vision.
It implies that AUG could become a widely adopted technique for efficient domain adaptation.

**Weaknesses:**

## Limited Base Model Evaluation
This paper uses OpenLLaMA v2 (3B) as the sole foundation model for all experiments. While this choice supports reproducibility and controlled distribution shift analysis, it limits the generalizability of the results.
AUG is designed to be broadly applicable, but its effectiveness on stronger, more widely used models (e.g., LLaMA 2/3, Mistral, Falcon, GPT-family) remains untested.
To strengthen the claim of scalability and robustness, future work should include experiments on larger and more diverse foundation models.

## Lack of Theoretical Guarantees
Although the current paper presents a well-motivated gradient formulation and matrix augmentation strategy, it does not provide formal theoretical analysis regarding convergence, generalization bounds, or robustness under distribution shift.
This weakens the depth of the contribution from a theoretical standpoint.
AUG introduces memory overhead due to conditioning on representations derived from frozen parameters, which may require optimization strategies such as checkpointing or selective augmentation.

## Memory Efficiency Trade-offs
AUG introduces additional memory overhead due to gradient computation involving frozen parameters. While this is acknowledged, the paper does not explore strategies to mitigate this cost (e.g., gradient checkpointing, selective layer augmentation). A comparative analysis of memory vs. performance trade-offs across PEFT methods would help practitioners make informed decisions.

## Limited Hyperparameter Optimization Transparency
The paper states that hyperparameters (learning rate, token length, batch size) were independently tuned per task, but it does not clarify whether this tuning was also done per PEFT method. This raises concerns about fairness in performance comparisons. Including details on the tuning protocol or reporting sensitivity analyses would improve transparency and reproducibility.

## Narrow Scope of Baseline Selection
While the paper compares AUG against several PEFT methods (BitFit, IA3, LoRA, AdaLoRA), it omits other relevant approaches such as QLoRA (Dettmers et al., 2023), Prefix Tuning (Li & Liang, 2021), and Prompt Tuning (Lester et al., 2021). These methods are widely used and could offer complementary insights, especially in low-resource or few-shot settings. Including or at least discussing these baselines would strengthen the empirical foundation.

## SAR Task Evaluation Scope
The SAR interferometry task is novel and compelling, but its evaluation is limited to a single region and time frame. Broader geographic and temporal validation would help assess the robustness of the method in real-world remote sensing applications.

**Questions:**

## Q1. Could you provide results using more recent or stronger foundation models (e.g., LLaMA 2/3, Mistral, Falcon)?
The current experiments rely solely on OpenLLaMA v2 (3B), which is relatively lightweight and dated. Since AUG depends on the quality of frozen parameters, its performance may vary significantly with stronger models. Demonstrating robustness across modern LLMs would greatly enhance the generalizability and practical relevance of the method.

## Q2. How does AUG perform in few-shot or in-context learning settings compared to prompt-based PEFT methods like Prefix Tuning or Prompt Tuning?
These methods are widely used in low-resource scenarios. Including them in the comparison or discussing their limitations relative to AUG would help contextualize the contribution more clearly.

## Q3. Can you clarify whether hyperparameter tuning was done independently for each PEFT method, or only per task?
This paper mentions task-level tuning but does not specify whether each PEFT method received its own optimized configuration. This affects the fairness of comparisons and should be clarified.

## Q4. Have you considered strategies to reduce the memory overhead introduced by gradient conditioning on frozen parameters?
AUG’s memory usage is higher than other PEFT methods. Exploring techniques like selective layer augmentation, gradient checkpointing, or partial conditioning could improve its practicality.

## Q5. Could you elaborate on the theoretical implications of gradient conditioning in AUG?
While the current paper presents a compelling intuition and formulation, it lacks formal analysis of convergence or generalization. Even partial theoretical insights or empirical ablations isolating the effect of conditioning would strengthen the method’s foundation.

## Q6. How sensitive is AUG to the choice of augmentation width (d_aug)?
This paper presents results for AUG 2 and AUG 4, but a more detailed analysis of how performance scales with augmentation size would help guide practical deployment.

## Q7. Is the SAR interferometry task generalizable to other regions or sensor types?
The task is novel and impactful, but the evaluation is geographically and temporally narrow. Clarifying its extensibility would help assess its broader significance.

## Q8. Do you plan to release code or pretrained models to support reproducibility?
The proposed method appears reproducible based on the descriptions, but public code would facilitate adoption and further validation.

---

> ### Author Response · Authors · 2025-12-01
>
> Q1) We choose open-llama v2 because it is guaranteed that it is only pretrained on English datasets. This provides a controlled testing environment compared to more recent LLMs
> Q2) We find few shot and prompt based approaches to be out of scope for our work. These methods do not work for high distribution shifts
> Q3) Hyperparameter tuning was done independently for each PEFT method for each task
> Q4) We welcome this idea, thank you.
> Q5) We note the reviewer's comment
> Q6) Performance improves with d_aug. We will include the analysis in the next version.
> Q7) Yes, the task is generalization. Figure 4 only shows an excerpt of the results for qualitative analysis while Figure 3 shows the aggregate results.
> Q8) We plan on releasing code and data

---

### Official Review · Reviewer_tEQn · 2025-10-28

**Soundness:** 2
**Presentation:** 1
**Contribution:** 2
**Rating:** 4
**Confidence:** 4

**Summary:**

The paper investigates a Parameter-Efficient Fine-Tuning (PEFT) approach for domain adaptation, with a focus on handling distribution shifts. While the proposed method demonstrates competitive performance on certain tasks, there are significant concerns regarding experimental setup, the clarity of citation formatting, and the claimed advantages of the proposed approach over existing methods such as LoRA.

**Strengths:**

1. The introduction of a new PEFT approach tailored to domain adaptation tasks is an interesting contribution.

2. The empirical results, particularly for low-resource language tasks, suggest that the proposed method can outperform existing PEFT techniques in certain settings.

**Weaknesses:**

1. Improper citation format: The paper does not follow standard academic citation conventions. Most references are presented in the author–year format (e.g., “Hu et al. (2021)”) rather than numerical or bracketed styles, which significantly affects readability and professionalism.

2. Limited experimental improvement: Compared with LoRA, the proposed method shows no significant performance gain under several experimental settings (see Table 1). The results fail to convincingly demonstrate the claimed advantages.

3. Increased parameter count: The proposed approach introduces additional fine-tuning parameters, which contradicts the premise of parameter efficiency and undermines the motivation of this work.

**Questions:**

1. Could you clarify the computational complexity and parameter increase introduced by the proposed method compared to LoRA in Table 3?

2. How do you justify the trade-off between parameter efficiency and the increase in fine-tuning parameters in your approach?

---

> ### Author Response · Authors · 2025-12-01
>
> 1) We note the reviewer's citation concerns
> 2) The reviewer references table 1 and claims that there is no significant performance gain. For English tasks, we find that PEFT methods perform similar to each other. However, the performance increases are we investigate multilingual (+2%), multimodal(+6%) and multitemporal settings (+3%).
> 3) It is not unusual for PEFT methods to increase parameter counts. For example LORA introduces new parameters in the low rank space and additively absorbs the new parameters at inference. To our knowledge, only Bitfit does not result in increased parameter counts because the bias parameter is already present.
>
> Questions
> Could you clarify the computational complexity and parameter increase introduced by the proposed method compared to LoRA in Table 3? Inference speeds are similar across PEFT methods. The parameter increase for AUG_2 setting is similar to LORA with R=4 while AUG_4 is similar to LORA R=8.
> How do you justify the trade-off between parameter efficiency and the increase in fine-tuning parameters in your approach? It is unclear what "increase in fine-tuning parameters" is referring to, since we show that we only finetune 0.12% of the parameters. This is comparable to LORA and ADALORA and fits the definition of PEFT fine tuning i.e. tuning less than 1% of the frozen weights.

---

### Official Review · Reviewer_1LHt · 2025-10-31

**Soundness:** 1
**Presentation:** 2
**Contribution:** 4
**Rating:** 2
**Confidence:** 4

**Summary:**

This paper aims to propose a new approach for parameter-efficient fine-tuning for a foundation model. They propose to augment the pre-trained model's weights by training a newly introduced lightweight matrix. Additionally, they propose to apply their approach to SAR satellite imagery to predict the presence of charcoal production kilns. Their empirical results indicate that in some cases, their proposed approaches can outperform several parameter-efficient tuning techniques.

**Strengths:**

1. Overall, their writing is clear and easy to follow.
2. Their ideas of exploring PEFT techniques in different domains should be interesting to many readers.

**Weaknesses:**

1. They lack a methodological comparison to existing work, representation fine-tuning [1]. I guess this work should be the extreme case of ReFT.
2. Also, their formulation is actually very similar to Lora. Eq.3 in [2] is Eq. 3 in this submission. Considering this, this work lacks novelty in its methodology.
3. They lack enough experimental results for the analysis of their approach and existing ones. They should show the results of varying the number of model parameters to tune and see the trade-off.
4. Their analysis is a bit superficial. They state that " In this work, we investigate how PEFT methods scale with varying degrees of distribution shifts and propose a novel PEFT method designed for domain adaptation." in abstract. However, their empirical analysis on domain adaptation for language models is performed on just one setting without investigating factors that can influence the performance difference.


[1] ReFT: Representation Finetuning for Language Models

[2] https://arxiv.org/pdf/2106.09685

**Questions:**

Please respond to my concerns about the weaknesses. I think all questions are important ones to address.

---

> ### Author Response · Authors · 2025-12-03
>
> 1) We acknowledge the reviewer's comment and will benchmark against REFT
> 2) The formulation is a generic affine transformation common for neural networks, however, the parameters of the transformation between LORA and our method are different e.g. LORA parameters for the formulation are a product of matrices BA which are summed elementwise in the forward pass while our formulation parameters are frozen and trained weights in the backward pass.
> 3) We acknowledge the feedback
> 4) It is inaccurate for the reviewer to state that we only test on one setting. The LLM we use has only been been pretrained on English text. Therefore, multilingual, image captioning (multimodal) and sensor timeseries (multimodal & multitemporal) are indeed different settings. Maybe the reviewer can clarify why these are not considered disparate settings

---

### Official Review · Reviewer_3ouD · 2025-11-01

**Soundness:** 2
**Presentation:** 2
**Contribution:** 2
**Rating:** 2
**Confidence:** 5

**Summary:**

This paper investigates how Parameter-Efficient Fine-Tuning (PEFT) methods perform under varying degrees of distribution shifts, proposing a novel PEFT approach called matrix vector product augmentation for domain adaptation. The method augments pretrained weights with learnable parameters, conditioning updates on both inputs and frozen knowledge, and is evaluated across English, multilingual, multimodal, and remote sensing tasks. Results show it outperforms state-of-the-art PEFT methods in low-resource and multimodal settings while remaining competitive on in-domain tasks, though it requires more memory than some alternatives.

**Strengths:**

1.  The proposed method achieves superior performance on tasks with large distribution shifts (e.g., low-resource languages and multimodal benchmarks) compared to other PEFT techniques.

2. It maintains parameter efficiency by tuning less than 1% of the model parameters, reducing computational costs while matching or exceeding full fine-tuning in certain scenarios.

**Weaknesses:**

1. My first concern is that this method has very high memory usage and computational overhead. The proposed matrix vector product augmentation method consumes significantly more GPU memory compared to other PEFT methods. For instance, AUG 2 and AUG 4 require 39.1 GB of training memory, while BitFit uses only 27.8 GB. This indicates inefficiency in resource utilization, which could limit practical deployment in resource-constrained environments. The method's design, which references frozen parameters in the gradient computation graph, contributes to this overhead. Improving memory efficiency, such as by optimizing parameter interactions or incorporating compression techniques, would enhance scalability.

2. The method introduces augmenting parameters via concatenation with frozen weights, but it primarily validates on tasks with incremental distribution shifts. The paper notes that the approach conditions on both inputs and pretrained parameters , yet it does not thoroughly address how it handles extreme distribution shifts beyond the tested scenarios (e.g., domains with entirely unseen modalities). Expanding the theoretical foundation to include robustness guarantees for broader shifts would strengthen the method.

3. The experiments exclude the Fourier Transform method from most benchmarks due to its poor performance and long training time. This omission reduces the comprehensiveness of the comparison, as it prevents a full assessment of how the proposed method fares against all contemporary PEFT techniques. Including such methods, even with their limitations, would provide a more balanced evaluation.

4. For the SAR interferometry task, the positive-to-negative label ratio is 1:2, and oversampling is used during training. However, the paper does not discuss potential biases introduced by this imbalance, such as overfitting to majority classes or reduced model reliability in real-world deployments where kiln distributions may vary. Incorporating techniques like stratified sampling or adversarial validation could improve robustness.

5. The dynamic normalization introduced for multimodal data (e.g., SAR embeddings) improves adapter-based methods but harms low-rank methods like LORA. The paper only hypothesizes that normalization makes data indistinguishable in the low-rank space without empirical validation (e.g., through ablation studies or sensitivity analysis). Deeper investigation into why normalization fails for low-rank methods would clarify its limitations and guide future adaptations.

6. While the experiments cover tasks from English benchmarks to multimodal SAR data, the shifts are constrained to specific domains (e.g., multilingual tasks focus only on a few languages). The paper states that tasks progressively introduce larger distribution shifts , but it does not include extreme cases like entirely unseen languages or sensors. Expanding to more diverse datasets would better test the method's scalability.

7. The SAR results show performance variations under different conditions (e.g., vegetation cover), but the evaluation relies on a small sample (e.g., Figure 4 displays limited true/fpositive cases) and does not include cross-region validation. This limits insights into generalization; for instance, the model's performance on unseen geographical areas remains unverified. Incorporating larger-scale testing or transfer learning across regions would address this.

**Questions:**

See Weaknesses.

---

> ### Author Response · Authors · 2025-12-01
>
> The authors thank the reviewer for their excellent feedback.
> We agree with the issues raised and will work on addressing them in the subsequent draft versions.

---

### Note · Authors · 2025-12-31

I have read and agree with the venue's withdrawal policy on behalf of myself and my co-authors.